# Histories of Science Communication

Kristian H. Nielsen 

Centre for Science Studies, Aarhus University, 8000 C Aarhus, Denmark; khn@css.au.dk

**Abstract:** Science communication has been central to our understanding of Modern Europe, and it also plays an important role in other parts of the world. The aim of this article is to present key narratives—histories—about the development of science communication in Modern Europe and beyond. Surveying key contributions, the article identifies two main narratives about science communication in Modern Europe: one about widening gaps between science and the public (informational, epistemological, and moral gaps) and one about building bridges through dialogue, engagement, and participation. Beyond Modern Europe, the same narratives appear but often with important twists. The discussion about science communication in Latin America, for example, includes colonial and postcolonial dimensions, whereas the narrative about science communication (science popularization) in China emphasizes the embeddedness of science communication in national politics. Together, the histories show that science communication is not the diminutive or distorted form of science but rather the sum of social conversations around science.

**Keywords:** science communication; science history; information deficit; science–society dialogue; public participation in science; science popularization

---

## 1. Introduction

In May 1980, a small delegation of four American science communicators and science administrators, dispatched by the American Association for the Advancement of Science (AAAS), visited China to explore emerging arenas for science popularization (the preferred Chinese term for science communication, see Section 3.2.) (Kirsch 1981). The China Association for Science and Technology (CAST) hosted the delegation, treating the delegates to science film screenings, museum tours, and visits to major television stations in Beijing and Shanghai broadcasting science documentaries, public health programs, and science shows for children. The delegates were duly impressed with the scope of science popularization in China and the seriousness with which CAST tried to reach the "lost generation", i.e., those whose training and education had been lost during Chairman Mao's Cultural Revolution. Jeffrey W. Kirsch, Director of the KPBS-TV Science Center at San Diego State University and member of the delegation reflected after the trip: "I can't help but feel privileged to have observed one aspect of a national society in transition" (Kirsch 1981, p. 35).

In their report to the AAAS Board of Directors, the delegation rehearsed two tropes familiar to science historians and science communication scholars. They coupled China's overall interest in science popularization with "an awareness of deficits and a determination to catch up on all fronts" (Kirsch 1981, p. 31). The delegation, like their Chinese hosts, obviously perceived a gap between science and the public in terms of knowledge and competencies, which needed to be filled by means of dedicated science popularization efforts. Although appreciative of the dedicated attempt to reach out to large audiences, the delegation lamented the authoritative approach to science popularization with rigid control of topics and narratives. In China, Kirsch concluded, "nothing is independent of politics and science popularization is no exception" (Kirsch 1981, p. 35).

---

## 2. Science Communication in Modern Europe

The AAAS delegation, much like their Chinese hosts, placed Chinese science popularization efforts in the context of history, nation building, media, and politics. This is also the task of historians of science popularization or science communication, to use the term more widely used today. Historians generally seemed to have confirmed the conclusion reached by Jeffrey Kirsch and the AAAS delegation that science communication deserves recognition because historical actors since the Enlightenment (and even before) have valued science communication as being useful and appropriate in relation to envisaged social or cultural transitions (Muñoz Morcillo and Robertson-von Trotha 2020). Situating science communication in historical contexts, historians (and science communication researchers) have produced at least two different narratives about the embeddedness of science communication in society and culture. Together, these narratives—one about gaps and one about bridges—emphasize the importance of science communication for situating science at the heart of our Modern World (Shapin 2007).

### 2.1. Science Communication to Fill Widening Gaps

In the early nineteenth century, science communication was useful to social reformers seeking to promote (or alleviate) industrial change by informing broader segments of society about advances in science and technology (Broks 2006). Various institutions, such as the Mechanics' Institute with branches in Great Britain and across the vast British Empire, endorsed the view that science and education were necessary for economic progress as well as social stability in industrialized societies. Various forms of popular science prepared workers and the emerging middle classes for their technological present and future, while also serving to build professionalism in the scientific community by way of perceived epistemic boundaries between scientific experts and laypersons. Due to increasing demand for scientific education and entertainment, the marketplace for popular science expanded (Fyfe and Lightman 2007), and scientists, science mediators—such as science writers, illustrators, and publishers—and the public all believed that there were "widening gaps" to be filled to mobilize the economic and political potential of industrialized mass societies (Broks 2006, p. 59).

The notion of widening gaps became even more pronounced in the early twentieth century, partly due to the deflation of the notion of opinion and partly due to the esoteric nature of increasingly specialized and mathematized scientific disciplines (Bensaude-Vincent 2001; Broks 2006). Three gaps could be discerned: an informational gap in terms of how much more scientists knew about certain topics; an epistemological gap to distinguish scientific reasoning from mere opinion; and a moral gap between scientists, committed to truth-seeking by way of universality, objectivity, and openness, and the rest of society seeking power, politics, and profit. Moreover, there was a perceived gap in terms of lacking appreciation of what J. D. Bernal in 1939 called the social function of science (Bernal 1939). Scientists and other (socialist) members of the social relations of science movement, including a few science journalists such as B. Michelsen, who after World War II served as Head of UNESCO's Division for Science & Its Popularisation, informed wide audiences about the benefits of science for society. They also encouraged public support for science and the use of science in a centrally planned society (Nielsen 2008; Werskey 1978).

During the Cold War, when scientists and policymakers on both sides of the Iron Curtain emphasized the significance of science and technology for prosperity and national security, the narrative about widening gaps found expression in calls for public appreciation of science, scientific literacy, and public understanding of science (DeBoer 2000; Lewenstein 1992; Locke 2002). Communicating science implied more than the transmission of scientific information to fill specific deficits; it served to change public attitudes to science, i.e., to overcome an "attitudinal deficit" (Bauer and Falade 2014, p. 149). Even though public surveys in many countries show persistent trust in science and support for science (Bauer and Falade 2014; Krause et al. 2019), scientists and science communicators have raised concern about diminishing public trust in science and promoted more and improved science

communication. Famously, the Royal Society's 1985 report on the public understanding of science found "[h]ostility, even indifference, to science and technology," threatening to undermine public understanding of science and, by proxy, the nation's industry (Royal Society 1985, p. 9).

### 2.2. Science Communication to Build Lasting Bridges

One way to address the problem of "widening gaps" is to fill the gaps. Building bridges is another way through dialogue, engagement, and participation. The Royal Society's 1985 report placed responsibility for the public understanding of science on the scientific community, arguing that "[s]cientists must learn to communicate better with all segments of the public, especially the media" (Royal Society 1985, p. 24). Critics claimed that the public-understanding-of-science agenda espoused paternalism and scientism (Bucchi 2008). The "new mood for dialogue", detected in 2000 in the UK House of Lords report and supported by wider calls for citizen participation and knowledge coproduction, particularly at the European level, invited public engagement with science, even public participation in science (Bucchi 2008, p. 67). To make science part of democratic deliberations and enable knowledge exchange between scientific institutions and civic society, many initiatives and programs have aimed at connecting publics—imagined and constructed as concerned citizens—to scientists and scientific expertise. The citizen science and the participatory technology assessment movements involve members of the public in cocreation of scientific knowledge and decision-making, respectively (Einsiedel 2021).

While science communication activities since the 1990s increasingly have been aimed at building bridges between society and science, historians of science in the same period have tried to reconstruct the history of popular science in terms of interactions between science and popular culture (Cooter and Pumfrey 1994; Topham 2009). They took inspiration from sociologists of science such as Richard Whitley who already in 1985 argued that there were no barriers or gaps, but rather cultural proximity between "the producers of scientific knowledge and the general laity" (Whitley 1985, p. 7). Historian of science James Secord, for example, placed Charles Darwin's evolutionary theory firmly in the context of Victorian culture, arguing that "what once made sense as the 'Darwinian Revolution' must be recast as an episode in the industrialization of communication and the transformation of reaching audiences" (Secord 2000, p. 4). In effect, science communication activities, whether specialized communication among experts or communication aimed at wider audiences, are central to our understanding of scientific developments as well as cultural appropriations of science. Secord, in a seminal 2004 essay, proposed a new narrative framework for historians of science, focusing on "processes of movement, translation, and transmission" with the potential for "a more effective dialogue with other historians and the wider public" (Secord 2004, p. 654).

More recently, prominent historians of science, such as Oreskes (2019) and Shapin (2007, 2019), have taken up Secord's challenge to engage in dialogue with wider audiences. In the face of spreading misinformation and rising doubts about scientific knowledge, often facilitated by organized campaigns seeking to discredit science, they have defended science by explaining how science relates to society and why the public should trust science. Oreskes drew on feminist philosophy of science to argue that the trustworthiness of science stems from its many social and institutional aspects, such as consensus, diversity, methods, reliance on evidence, and values. According to Oreskes (2019), science should be trusted because it is a collective enterprise organized around the production of certified and reliable knowledge. Shapin (2007) has drawn attention to the bridges between science and society built over the twentieth century. The problem we confront, he says, "is better described not as too little science in public culture but as too much" (Shapin 2019). As historians of science and science communication, we should not see science as an external force shaping or impacting on our modern world, but rather pay attention to the taken-for-granted and often invisible ways in which science has become gradually embedded in policy making, innovation, security, industry, and other social institutions of power and wealth creation.

### 3. Science Communication beyond Modern Europe

The historical narrative about science, science communication, and modernity sparked ideas about building bridges across gaps in the European context and stories about fighting superstition with science in North America (Schirrmacher 2013). Recent contributions to the field of science communication research have shed light on conceptions and histories of science communication in other countries and regions (Gascoigne et al. 2020; Schiele et al. 2012; Schiele et al. 2021). These efforts reveal shared ideals in terms of fostering a science culture with international aspirations, but also historical and cultural differences.

*3.1. Science Communication in Latin America: Social Appropriation*

For example, the emergence of science communication activities in 19th century Latin America took place in a colonial or the immediate postcolonial context. In Brazil, the first science communication initiatives seemed to have been affiliated with Portuguese institutions established after the transfer of the Portuguese Court to Rio de Janeiro in 1807. After the declaration of independence in 1822 and intensified by the establishment of a direct submarine telegraph link to Europe in 1874, Brazilian newspapers and magazines reported on scientific advances to elite audiences. Later, in the 1920s, the newly established Brazilian Academy of Sciences used science communication as a tool to promote national scientific institutions with an emphasis on basic and not applied science (Massarani and Moreira 2016). Recent developments in Latin American countries with more advanced research and innovation systems, such as Argentina, Mexico, and Brazil, have led to increased attention to science communication shaped by conflicting forces. In these settings, science communication is enfolded in a tradition of top-down policy making, great cultural diversity, strong social movements, especially in fights for indigenous rights, democratization, and the environment, and little involvement of the private sector in research and development activities (Polino and Castelfranchi 2012).

The growing interest in many Latin American countries led to the formation of new ideas about science communication. In Colombia, the acronym ASCyT (Apropiación Social de la Ciencia y la Tecnología, Social Appropriation of Science and Technology) has been used to show that science communication is (not only) about the dissemination of scientific content to the public, but rather involves complex interactions between science and culture. ASCyT takes inspiration from Euro–American discussions about gaps vs. bridges, cf. Section 2. Moreover, it implies a new historiography of science communication in Latin America, which is sensitive to cultural differences and specificities or to paraphrase James Secord, processes of movement, translations, and transmission of conceptions and practices of science communication across cultural settings. This new historiography would therefore seem more appropriate to a historical understanding of science communication in global settings, including Europe and North America (Daza-Caicedo et al. 2020).

*3.2. Science Communication in China: Popularization and Politics*

In China, the dominant phrase used to describe science communication activities and histories has been and still is science popularization (*kepu*, 科普) (Ren 2019; Yin and Li 2020). There are numerous historical examples of learned scholars trying to communicate Western science by way of translation, for example Matteo Ricci, the Italian Jesuit priest who arrived in China in 1582 where he initiated the translation of Euklid's *Elements* and many other scientific works into Chinese (Zhu 2017). Science popularization took on importance in the Chinese republic established in the early twentieth century with science being taught in schools and science (and science popularization) generally being a way to rebuild China. The founders of the Communist party, such as Chen Duxiu, originally saw a close connection between science and democracy, but as China's political institutions during Mao Zedong became more authoritarian and oppressive, science was severed from democratic values and affiliated only with technological and economic advances (Eiterjord 2018).

The Chinese word for science, *keji* (科技), means both science and technology. The conflation of the two terms indicates that science in China is valued for its societal and commercial applications, which again are defined by government (Eiterjord 2018). Science popularization too is highly institutionalized and—as Kirsch (1981) observed in China in 1980—politicized. The political reforms initiated by Deng Xiaoping from 1978 onward placed science and technology at the core of Chinese society as "productive forces" and sparked new interest in science popularization (Yin and Li 2020, p. 211). The many initiatives and activities, although often pursued by scientific elites in a top-down manner and for the general good of the economy, have now "merged with the social life of Chinese citizens" (Yin and Li 2020, p. 217). In addition, inspired by Western developments described earlier, but also informed by rapid social transformations in China, scholars have begun to criticize science popularization, arguing that the Chinese science popularization model needs to be replaced with a more dialogue-oriented science communication model (Ren et al. 2012).

## 4. Conclusions

When Jeffrey Kirsch and the AAAS delegation in 1980 visited China to explore emerging topics in science popularization or science communication, the history of the field was also just emerging. Histories of science communication have been written by historians of science and by scholars and practitioners involved in actual science communication or science communication research. There are key topical, temporal, and regional variations in the historiography of science communication. Science communication includes everything from popular science specifically produced to disseminate scientific knowledge (or knowledge about how science works) to cultural products, political statements, and social interactions with a science content. Science communication, in the words of science communication scholars Massimiano Bucchi and Brian Trench (Bucchi and Trench 2021), is "the social conversation around science". Science communication therefore evolves in specific historical and geographical contexts. It has been central to the development of the idea of modernity that arose in Europe during the 19th century, but also in colonial and postcolonial contexts and in authoritarian regimes. Science communication continues to have great significance for our understanding of past, present, and future relations between science and society.

**Funding:** This research received no external funding.

**Institutional Review Board Statement:** Not applicable.

**Informed Consent Statement:** Not applicable.

**Data Availability Statement:** Not applicable.

**Acknowledgments:** The author would like to thank three anonymous reviewers for useful comments.

**Conflicts of Interest:** The author declares no conflict of interest.

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
