# Peer review of "Histories of Science Communication"

_2409-9252, doi:10.3390/histories2030024_

Round 1

Reviewer 1 Report

This is a fine brief account of science communication in Europe, China and Latin America which, clearly, cannot do justice to the vast literature. Unfortunately, Europe (and the US) is not differntiated with respect to diverging politically and culturally defined approaches (cf. Science in Context Vol. 26/3 (2013) "Communicating Science: National Appoaches in 20C Europe"; Burnham: How Superstition Won ..., 1987). The title and its equating of "key narratives" with "histories" may be reconsidered.

Author Response

In the revised manuscript, I have made explicit differences between the European and North American context (p. 3, l. 132-133). I suggest keeping "histories" in the title unless the editors think it is better to use "key narratives". 

Reviewer 2 Report

Review of ‘Histories of science communication”

1. The paper is not about a new theme, as the History of science communication has been surveyed by multiple authors. The Author intends to present a review of the literature concerning two main key narratives. These two narratives aren’t also new.

2. The most original point of view presented is the look beyond Europe, as it is presented in the Abstract. Keywords are ok.

3. Introduction is not correct. It does not present the starting question, the objectives of the paper, the context, the motivation, or the paper structure. It should be improved to address those issues.

4. Section 2 presents the gaps and bridges narratives also known in the literature as the deficit model and the dialogue model, but I think it misses the public engagement and citizen science most recent perspectives, which goes beyond the dialogue setting. It also misses the technological and information changes, and the scientific disputes, e.g. the climate change debate, as the public demands new forms of scientists’ behavior, and a strong role of science against pseudoscience and disinformation phenomena. It’s not discussed the relationship between the two narratives. In fact, some authors establish an evolving perspective between them. Other depict their coexistence. I’ve found Section 2 a weak piece of text which should be better sustained.

5. Section 3 tries to endorse a global perspective of science communication, but it is also scarce, with only a few paragraphs. The paper sometimes seems a draft to produce a larger one.

6. In the conclusion, the Author claims that there are variations in the historiography of science communication, but where are they in the paper? Later, the Author writes that science communication ‘evolves in specific historical and geographical contexts’, but this would have needed a larger development and a more thorough and comprehensive literature review.

7. In the end of the conclusion, the Author argues that science communication ‘has been central to the development of the idea of modernity that arose in Europe in the course of the 19th century, but also in colonial and post-colonial contexts and in authoritarian regimes.’ This is not explained in the paper, but this sentence could be the main argument or hypothesis to be developed afterwards. This paper again fails to address this issue.

8. The paper doesn’t have a methods section, which is mandatory, in my view.

9. The paper is correct, but it should be reinforced to produce a better explanation of the Author’s arguments. It’s not a literature review but a few topics poorly aligned.

Author Response

The reviewer seems to have missed the fact that this paper is not an original research article but submitted for a special issue with very specific requirements: short essays (max. 24.000 characters) aimed at presenting history of science topics to general historians. I believe that I have shown (or at least made visible) "topical, temporal, and regional variations in the historiography of science communication": gaps/bridges/appropriation/transit/politics, historical developments since the 19th century, and Europe/North America/Latin America/China.

Reviewer 3 Report

I started to read the manuscript "Histories of science communication" with a great interest, since the abstract promises potentially highly interesting inquiry connecting European and non-European takes on science communication. Such an approach is highly ambitious, given the wide scope and long history of science, and the variety of different and evolving contexts and actors involved. Therefore, I also expected to see either a very long manuscript or tight focus on selected cases serving as examples of larger developments.

Unfortunately, the manuscript largely fails to meet the expectations. In my opinion, the manuscript could be developed towards a perspective or viewpoint type of short contribution but currently it lacks clear critical insight typical for such contributions.

In my opinion, the manuscript does not meet the requirements needed for a research article. The very short text presents some examples around the world and from different time periods, but the examples are not really tied together. The manuscript is simply too short in relation to very wide approach. Clear research question is missing, theoretical rooting and analytical approach is not explained, and empirical material appears to be rather random selection of examples around the world. Furthermore, after reading the manuscript I kept wondering what were the novel results from the study.

The theoretical background and empirical materials of the study are not explained. Secord's knowledge-in-transit framework is briefly mentioned in the middle of the text. Apparently, this framework was considered unsuitable for studying European developments, but reasons for not using this framework to guide the whole study is not opened up and it remains unclear how it was employed to interpret the non-European developments.

More careful use of previous literature is needed. For example, the claim (lines 35-36) about "the two tropes familiar to science historians and science communication scholars" is made without references. This leaves open whether the claim is about all historians and scholars or e.g. scholars of modern European times?

The term science communication is not properly defined, only referred in relation to popularisation (that is not defined either).

The criteria for selecting examples are not explained. I feel that too wide interpretations are made based on isolated examples. I’m not entirely convinced about using examples of activities of individual persons to justify very broad claims (such as B. Michelsen  in line 83).

The distinction between “widening gaps” and “building bridges” is the backbone of the current analysis. However, more detailed description of "widening gaps" needed. There should be more sensitiveness to (potential) national differences and identification of similarities. For example, the paragraph about the Cold War era could specify in more detail the similarities of science communication or popularisation in very different policy systems. The section 2.2. does not coherently connect "widening gaps" and "building bridges" approaches. The examples selected to illustrate building bridges -approach seems to lack connection with wider conceptual, temporal or spatial developments. I suppose the aim here should be to give an overall picture of the “building bridges” -approach in modern Europe? The section focusing on Latin America lacks detail and should be left out from this study.

I hope my evaluation is not too harsh, but I am quite convinced that the study is simply too ambitious and aims to cover far too wide issues in overly concise form. Overall, there is a need to build the study around carefully selected tight focus that helps to narrow down the wide cross-cultural comparison in a meaningful way.

Author Response

The reviewer seems to have missed the fact that this paper is not an original research article but submitted for a special issue with very specific requirements: short essays (max. 24.000 characters) aimed at presenting history of science topics to general historians. I agree with this reviewer that Secord's framework is not only applicable to non-European contexts. I have therefore explicitly mentioned that Secord's framework - taking inspiration from the Latin American case - is appropriate to a a historical understanding of science communication in global settings, including Europe and North America (p. 4, l. 163-164).

Round 2

Reviewer 2 Report

Despite the paper requirements, I think that the Authors ignored all my recommendations, specially the missing point of view on section 2, as I have already mentioned in my first review. For that reason, I maintain the decision to reject.

Author Response

In response to the reviewer's comment, I have added a paragraph to section 2. The section now includes historical reflections/reactions to the current challenges faced by science and science communication as suggested by the reviewer.

Reviewer 3 Report

Thank you for the revised version, and for clarifying the type of the contribution. As a perspective type of essay I am willing to accept the revised manuscript as I welcome attempts to bridge different perspectives to science communication.

Author Response

Thank you for accepting the revised essay.

Round 3

Reviewer 2 Report

The Author maintains the decision to not answer to all the issues presented in my review report.